# Assessing Forest Vulnerability to Climate Change Combining Remote Sensing and Tree-Ring Data: Issues, Needs and Avenues

Santain S. P. Italiano [1,*], Jesús Julio Camarero [2], Michele Colangelo [1,2], Marco Borghetti [1], Maria Castellaneta [1], Manuel Pizarro [2] and Francesco Ripullone [1]

[1] Scuola di Scienze Agrarie, Forestali, Alimentari e Ambientali, Università della Basilicata, Viale dell'Ateneo Lucano 10, 85100 Potenza, Italy; mcolangelo@ipe.csic.es (M.C.); marco.borghetti@unibas.it (M.B.); maria.castellaneta@unibas.it (M.C.); francesco.ripullone@unibas.it (F.R.)

[2] Instituto Pirenaico de Ecología (IPE-CSIC), Avda, Montañana 1005, 50192 Zaragoza, Spain; jjcamarero@ipe.csic.es (J.J.C.); m.pizarro@csic.es (M.P.)

[*] Correspondence: santain.italiano@unibas.it

**Abstract:** Forests around the world are facing climate change. Increased drought stress and severe heat waves in recent decades have negatively impacted on forest health, making them more vulnerable and prone to dieback and mortality phenomena. Although the term vulnerability is used to indicate an increased susceptibility of forests to climate change with a worsening of their vigour status that can compromise their ability to respond to further climate extreme events, there are still uncertainties on how to evaluate it. Indeed, evaluation of forest vulnerability is complex both because of some critical issues in the estimation methods used and because of the multiple factors influencing the response of forests to ongoing climate change. A way to assess the vulnerability to environmental stresses is by combining remote sensing and dendroecological data. However, these two approaches entail multiple uncertainties, including growth/photosynthetic relationships, carbon allocation dynamics, biases of tree-ring data and noisy remote sensing data, which require further clarification for proper monitoring of pre- and post-drought forest trajectories. Our review aims to create an overview of the current literature and knowledge to understand the critical issues, needs and possible solutions that forest vulnerability research is addressing. We focus on Mediterranean forests located in a climate warming hotspot and showing a high vulnerability to increased aridification.

**Keywords:** climate change; drought; dieback; forest vulnerability

## 1. Introduction

Average temperatures in Mediterranean regions have risen +1.4 °C since the late 19th century, an increase of +0.4 °C above the global average. Overall, these conditions are leading to a significant reduction in summer precipitation availability and more frequent heat waves and droughts [1,2]. Such climatic alterations can affect forest ecosystems by causing dieback and consequent changes in forest composition, structure and distribution, making some stands more susceptible to pests, pathogens [3] and fires [4]. Thus, overall, ongoing climate change makes our forests more vulnerable by compromising their ecosystem services such as $CO_2$ uptake, climate change mitigation, biodiversity conservation, soil protection, water regulation, etc.

Tree mortality phenomena, due to the physiological stress to which forests are subjected, have been attributed to interconnected abiotic and biotic stress factors. Hydraulic failure and carbon starvation, followed by attacks by biotic agents, have been identified as the main mechanisms of mortality [5].

Hydraulic failure, which occurs in hot and dry conditions, results from an imbalance between plant transpiration and water uptake, which causes an increasingly negative

xylem pressure, leading to widespread xylem embolism, interruption of water flow and canopy desiccation [5]. Carbon starvation, on the other hand, occurs because the plant, under water stress conditions, closes its stomata to limit water loss through transpiration, resulting in reduced $CO_2$ supply and carbohydrate synthesis [5]. Plants initially utilise stored carbohydrates to cope with stress [6]; these reserves cannot be restored if water stress conditions persist, and therefore the plant goes into starvation. Furthermore, drought negatively affects phloem transport with a reduction in cell turgor, preventing carbon translocation in the plant [7]. Therefore, the uptake of nutrients and minerals in the soil is reduced during dry periods, causing generalised stress [8]. Hydraulic failure and carbon starvation are mechanisms that can occur simultaneously [9,10] and can therefore trigger a reduction in photosynthetic activity (defoliation, canopy desiccation) and growth rate (very narrow growth rings) [11,12].

Climate can also influence insect, pathogen and disease cycles [13]. The combination of various factors, such as drought and excess nitrogen due to anthropogenic emissions ($NH_3$, $N_2O$, $NO_X$), causes a reduction in the concentration of allelochemicals in plant leaves [3], secondary metabolites produced by plants, which can have a defensive function against pests; the reduction of these substances can therefore make trees more susceptible to pathogens.

Thus, abiotic and biotic factors are interconnected and it is clear how climate change can trigger dieback phenomena and influence the vulnerability and growth of forests. Vulnerability can be defined as the degree of susceptibility of a system and its inability to cope with the adverse effects of climate change [14], or, more precisely, as the reduction/absence of response to such adverse climatic events (e.g., drought). In the case of forests, a loss of resilience or recovery capacity can result in forest dieback and higher mortality rates of trees, stands or entire forests [15].

Thus, heat waves and drought periods are causing the death of several forest tree species in most continents: North America, Europe, Australia, continental Asia and Russia, showing a sharp increase in mortality events from 1998 to 2000 [16,17]. In the Mediterranean basin, forest dieback phenomena have been reported in temperate oaks (*Quercus robur* L., *Qurcus petraea Matt. Liebl*) in northeastern Italy [18,19] and northern Spain [20], while in southern Italy, dieback has affected oak species such as *Quercus cerris* L., *Quercus pubescens* Willd. [21] and *Quercus frainetto* Ten. [22]. Other dieback phenomena have been reported in several conifers, such as *Pinus pinaster Aiton.* in southeastern Spain [23], *Pinus pinea* L. in northeastern Spain [24] and *Pinus sylvestris* L., *Pinus halepensis Mill.* and *Abies alba Mill.* in eastern Spain [25]. In the case of *A. alba*, the species showed a marked reduction in growth in some Pyrenean populations compared to cooler and wetter sites in eastern and central Europe [26,27]. Furthermore, several Greek stands of *P. sylvestris* L. and *Abies cephalonica Loudon* have shown dieback triggered by climate change and assisted by pathogen outbreaks [28]. In Greece, the increase in temperature caused mortality phenomena in *Pinus bruita* Ten., highlighting vulnerability to global warming [29], while *Abies borisii-regis Mattf.* manifested greater adaptability to increasing temperatures but susceptibility to drought periods [30]. In addition, climate change causes changes in the composition and distribution of species, leading to alterations in ecosystems. Recent studies [31] have shown, through analysis of different biomes around the world (131 sites), that tree mortality caused by drought phenomena led to a short-term (on average 5 years) conversion of vegetation type. Species self-replacement involved changes in community composition, e.g., from mesic forests to more xeric communities, with the spread of shrubs or non-woody vegetation. Thus, extreme drought could act as an environmental filter for species particularly sensitive to water deficit, highlighting a potential reorganisation of the ecosystem [31].

The phenomena of forest dieback are mainly manifested through symptoms such as crown defoliation and wilting, branch desiccation, crown decay, epicormic shoot production, longitudinal bark cracking, biomass reduction, necrosis of absorbing roots, growth decline, etc. [21,32]. Clearly, the assessment of forest vulnerability as linked to dieback is

not straightforward; this complexity is due to some critical issues in the estimation methods used and to the many factors that influence the response of forests to climate, including site characteristics (exposure, altitude, slope) [33], nutrient availability and soil type [8], species abundance [34], isohydric (water-sparing) vs. anisohydric (water-spending) strategies [35], the availability of reserves such as non-structural carbohydrates [7,36], stand structure [37,38], tree age [39,40] and mitigation or compensation processes [41], as well as management and human activities [42].

Therefore, the response of forests to climate change is influenced by several factors and variables, but since the repercussions of climate anomalies on forests are mainly translated into a reduction in radial increment and photosynthetic activity, the study of forest vulnerability can be conducted through dendroecological surveys, to analyse the increment of growth rings, and remote sensing vegetation indices, which provide information on the state of canopies (Table 1).

Field surveys and dendroecological/anatomical studies of wood in relation to climatic data [43] are fundamental; in fact, by analysing tree rings and anatomical variables of wood, it has been possible to understand that drought is the triggering factor of dieback in Mediterranean forests [21,25], while remote sensing surveys represent a widely used method to study forest dynamics over large areas [34,44] and to observe the response of forests to disturbances (drought, heat waves, fires) in spectral terms [45].

However, the methods used to estimate vulnerability may present some criticalities; for example, in some cases, the radial increment estimated with dendrochronology does not correspond to climatic dynamics, which may be due to the translocation of reserve carbohydrates used by woody plants to grow during periods of stress [46]. Or, in other cases, satellite information does not correspond to field observations. Indeed, coarse spatial resolution [47], cover mix (tree canopy, undergrowth, soil, etc.) [48] and density [49] or plant diversity [50], could lead to overestimated or underestimated spectral index values. Therefore, studying the vulnerability of forests to climate change is still an evolving challenge for researchers and survey methods need further implementation to understand how forests are able to respond to climate change and how resilient they are to such events. In this article, we provide an overview of the state of the art to reason about the studies and the main critical issues that have emerged regarding the combined use of tree rings and remote sensing to examine the vulnerability of forests.

**Table 1.** Recent articles with a combination of dendroecological and remote sensing approaches: MXD (maximum latewood density), TRW (tree-ring width), TRWi (tree-ring width index), BAI (Basal area increment), RWI (ring width index), GPP (gross primary production); VI (vegetation index): NDVI (Normalized Difference Vegetation Index), EVI (Enhanced Vegetation Index), NDWI (Normalized Difference Water Index).

| Reference | Species | Variables | Correlation |
|---|---|---|---|
| Moreno-Fernández et al., 2022 [24] | *Pinus pinea* L., 1753 | BAI, NDVI, NDWI | different responses to drought between indices and low BAI –VI correlation |
| Ogaya et al., 2015 [44] | *Quercus ilex* L., 1753 | Defoliation, BAI, NDVI, EVI | positive correlation between defoliation and vegetation indices |
| Coluzzi et al., 2020 [45] | Mixed forest of oaks and ash trees | Defoliation, NDVI | positive correlation |

**Table 1.** *Cont.*

| Reference | Species | Variables | Correlation |
|---|---|---|---|
| Brehaut et al., 2018 [47] | *Picea glauca* (Moench) Voss, 1907 *Picea mariana* (Mill.) Britton, Sterns & Poggenb., 1888 *Salix glauca* L., 1753 *Alnus crispa* (Aiton) Pursh, 1814 *Populus tremuloides* Michx., 1803, at different sites | TRW, NDVI | low correlation |
| Vicente-Serrano et al., 2020 [50] | 15 species in different biomes | TRW, GPP, NDVI | site-dependent relationships |
| Wang et al., 2021 [51] | *Pinus densiflora* Siebold & Zucc., 1842 | RWI, NDVI | positive correlation |
| Castellaneta et al., 2022 [52] | *Pinus sylvestris* L., 1753 *Quercus pubescens* Willd., 1805 *Quercus frainetto* Ten., 1813 *Juniperus phoenicea* L., 1753 | BAI, NDVI | positive correlations |
| Gazol et al., 2018 [53] | 11 tree species between gymnosperms and angiosperms | TRWi, NDVI | positive correlation |
| D'Andrea et al., 2022 [54] | *Picea abies* (L.) H.Karst., 1881 | RWI, NDVI | inconsistent trend |
| Lapenis et al., 2013 [55] | *Picea abies* (L.) H.Karst., 1881 | TRW, NDVI | inconsistent trend |
| Vicente-Serrano et al., 2016 [56] | 100 tree species in different biomes | TRW, NDVI | different relationships between growth and vegetation indices; stronger correlation in dry sites |
| Beck et al., 2013 [57] | Treeline vegetation mix in different forests | TRW, MXD, NDVI | positive NDVI-MXD correlation |
| D'Arrigo et al., 2000 [58] | *Picea glauca* (Moench) Voss, 1907 *Larix gmelinii* (Rupr.) Kuzen, 1854 | MXD, NDVI | good correlation |

## 2. Resilience Indexes to Assess the Vulnerability of Forests

As described above, the vulnerability of forests results in a loss of resilience, i.e., a reduced ability to return to pre-disturbance conditions, that over time can cause mortality phenomena [15]. Therefore, to assess forest resilience, resilience indices [39] can be applied based on dendroecological information, i.e., the performance of radial growth (e.g., BAI basal area increment) before (PreDr), during (Dr) and after (PostDr) drought episodes. These indicators are obtained through simple formulae and are represented by: resistance (Rt = Dr/PreDr), recovery (Rc = PostDr/Dr) and resilience itself (Rs = PostDr/PreDr). By applying these indices, it is possible to understand the dynamics of a forest stand

in response to stress episodes. Lloret et al. [39], analysing tree rings in *Pinus ponderosa Douglas* trees in remote Rocky Mountain forests, observed low radial growth correlated with drought periods, and estimating the components of Rt, Rc and Rs showed that impacts from a previous event and cumulative effects from the past, resulting in lower growth, caused a decrease in resilience [39]. Further studies conducted in Germany [35] applied resilience parameters [39] to quantify the growth response of trees to periods of water stress. Resistance, recovery and resilience (Rt, Rc and Rs) were estimated for three different species, *Fagus sylvatica* L., *Picea abies* L. and *Quercus petraea Matt. Liebl*, for both pure and mixed stands. In all cases, the species effect on Rt, Rc and Rs was significant; i.e., Norway spruce is easily affected by dry spells but recovers quickly and oak is more resistant but recovers more slowly, while beech's reaction is in the middle, showing greater resistance and resilience when mixed with oak than when monospecific [35]. Other studies [15] applied these indices (Rt, Rc, Rs) using a pan-continental tree ring width (TRW) database for entire biomes (118 sites and 3500 individuals) with different angiosperm and gymnosperm species. Overall, the TRW time series study showed reduced growth associated with higher mortality risk, and resilience indices showed that drought-dead trees were less resilient to drought events prior to their death than surviving trees for both gymnosperms and angiosperms.

Therefore, these indicators provide information on the resilience of forest stands, based on dendrochronological measurements, so even these indices could suffer from some criticalities linked to possible cases of inconsistencies between growth rings and climate [46,59,60], which we will discuss in the following sections.

## 3. Methods for Monitoring and Studying Forest Vulnerability

### 3.1. Tree Crown Evaluations

The analysis and monitoring of dieback and mortality phenomena have been addressed through various methodologies, such as visual analysis of vegetation conditions, field surveys, remote sensing techniques and many others. Some studies in the past have used a visual and qualitative assessment of trees (vitality classes) to evaluate the severity of dieback [61]. This approach consists of assigning each observed plant a vitality class, i.e., a numerical value in the range from 1 to 6 (healthy to dead plant) [45,61]. However, this method, being a visual and qualitative assessment of the state of the canopy, is not very objective, so it depends on the operator's ability to distinguish between the different vitality classes. Other studies [43,62], for example, have differentiated between declining and non-declining trees based on the current percentage of crown transparency or defoliation. This is a widely used practical approach to characterise tree vigour; nevertheless, this approach has been subject to some criticism. Indeed, establishing a fixed threshold of defoliation to distinguish trees in decline from those that are not can be questioned because crown transparency can change from year to year. Thus, a defoliated tree may recover, while some non-defoliated trees may start to die back. However, there are defoliation thresholds that, once exceeded, are not reversible.

Other studies have used remote sensing to assess forest cover. Indeed, it has been shown that airborne Lidar (Laser Imaging Detection and Ranging), through the acquisition of point clouds, can detect defoliation in terms of LAI [63] and thus provide feedback on canopy and forest vigour [64]. Given the complexity of forest systems and their response to disturbances, visual or remote canopy assessment methods must always be accompanied and validated by quantitative field surveys and measurements to ensure representativeness and correlation between the data obtained.

### 3.2. Dendroecology

Qualitative observations of canopies alone, therefore, are not sufficient to best discriminate the state of forests and, consequently, their vulnerability. Quantitative investigations to examine forest dieback phenomena can be obtained using dendroecological data; an example of this type of investigation is that employed by several studies [22,43], in which,

in addition to an initial visual assessment of canopy transparency, time series of tree rings were also obtained. Trees under drought conditions show a reduction in the radial increment and area of the vasal lumen and a consequent reduction in hydraulic conductivity [22]. Following frequent extreme weather events that trigger dieback phenomena, a decline in the growth of trees is observed long before their death, which can vary in intensity and duration. This phenomenon results in divergent growth trends between trees that experience dieback and those that do not [12,20]. Thus, the reduction in growth immediately before death could be due to a generalised water failure and/or secondary stress factors (diseases and pathogens) favoured by a loss of tree vigour, while a slow growth slowdown could be associated with a gradual decline in hydraulic performance and depletion of carbon reserves [65].

Therefore, tree rings and their anatomical variations are considered important proxies for studying the response of forests to environmental changes by retrospectively analysing, with high temporal resolution, the climatic dynamics permanently recorded in the wood structure [66].

However, even the growth ring does not always show reductions in growth during a particularly hot and droughty year, e.g., the formation of early wood in porous ring species depends on the remobilisation of stored carbon, thus not exclusively reflecting the climatic conditions during that actual growth period [46,60]. In other words, growth is maintained during drought through the use of stored carbohydrates, but this can cause depletion of non-structural carbohydrate (NSC) reserves and reduce the trees' resistance to further drought events, making them prone to death [15]. In addition, drought responses may vary depending on vegetative earliness, i.e., two species in the same area may show different growth responses depending on the time of sprouting, and thus the growth ring may or may not highlight the drought event [59].

In spite of these difficulties, to date dendrochronological surveys have been the most suitable for providing information and quantifying forest dieback phenomena, but these types of studies can only be applied to single sites on a small scale and require considerable resources, so even these alone do not allow for the study of large areas such as those affected by dieback.

### *3.3. Remote Sensing*

To obtain information on forest vulnerability on a large scale and save the time and resources needed for field surveys, remote sensing can assist. Indeed, satellite-based vegetation indices have made it possible to switch from individual- to forest-scale studies. Therefore, the combination of the dendrochronological approach and remote sensing is promising for assessing forest decline [51,67]. A widely used remote sensing index on which many other indices are based is the Normalized Difference Vegetation Index (NDVI) [68]. This index is widely used as a proxy for forest photosynthetic activity [45,51,52] and productivity in drought-prone Mediterranean biomes [50]. Thus, after drought events or heat waves, which lead to a reduction in photosynthetic activity, the NDVI tends to assume lower values, while higher values of the index indicate favourable conditions for plant health. Therefore, this index has been used to study mortality phenomena or increases in biomass related to climatic conditions [44]. For example, some studies [53] have shown the existence of a positive correlation between resilience indices [39], obtained using growth in terms of tree ring width indices (TRWi) and forest productivity in terms of NDVI. However, it must be remembered that the response of forests, as expressed by the remote indices, can differ depending on the type of forest site, tree species and the degree of stand mixing [69].

### 3.3.1. Decoupling of NDVI–Growth Relationship

NDVI is an index that measures photosynthetically active biomass (canopy of trees and greenness), but its relationship to growth is complex [47,70]. Indeed, changes in carbon allocation may favour foliage over woody biomass, leading to a weakening of the relationship between tree-ring growth and the remote sensing signal. This can cause

inconsistency phenomena between trends [54], i.e., the presence of positive NDVI trends when negative tree-ring trends are observed [55]. To study these relationships, Vicente-Serrano et al. [56] compared tree ring data with NDVI time series on a global scale, finding a high spatial and temporal divergence in forest growth responses. In fact, growth rates and vegetative recovery between coexisting species may differ and, respectively, carbon sequestration may vary and influence the growth of rings with respect to NDVI. Therefore, the different phenology of wood and leaf formation could explain the decoupling between NDVI and growth [56].

Other cases where NDVI may not provide an accurate account of radial growth are surveys in ecotone areas [57], i.e., at forest edges where there is a transition from tree to shrub vegetation, greening trends with increasing shrub biomass could alter vegetation indices. Furthermore, it has been shown that not only vegetation composition, but also slope [51], exposure and altitude, influence the climatic response, which means that in an area, different vegetation types or trends may confound the NDVI–ring width relationship.

Consequently, low image resolution, changes in resource allocation in trees and site characteristics may interact to limit the correlation between NDVI and annual radial growth [47]. These limitations increase with the complexity of the landscape, such as for highly heterogeneous Mediterranean ecosystems that manifest articulated responses to extreme events, so response patterns and tree-ring growth on NDVI time scales may not be fully representative [50]. On the other hand, if species composition is homogeneous or if the proportion of dominant species responds similarly to climatic variations, then there should be a positive correlation between NDVI and trends in ring width [58].

In order to use these two indicators (NDVI and tree rings) congruently, one could consider the observed positive correlations between MXD (maximum latewood density), NDVI and temperature during the growing season [57] and perform satellite analyses at a higher spatial and temporal resolution that could allow for a better investigation. Certainly, an examination of the relationship between NDVI and processes at the tree/species level, date of sprouting or root growth may lead to a better understanding of these dynamics [47]. Thus, appropriate research is needed to understand the physiological and phenological processes that explain the dependence between wood formation and photosynthetic processes underlying NDVI and the relative time intervals in which these processes occur [56].

In addition, the use of high-resolution satellite data could improve remote sensing information; in fact, Sentinel-2 10 m × 10 m space resolutions have given good results in small-scale monitoring [45] of the effects of extreme weather events on mixed Mediterranean forests in southern Italy, showing a good correspondence between NDVI and qualitative data collected in the field. To obtain highly detailed resolutions, an alternative could be remote sensing with drones, which allows lower material and operational costs and greater flexibility in spatio-temporal resolution than satellites [71]. Recent studies [72] have used unmanned aerial vehicles (UAVs) to monitor mixed coniferous and deciduous forests in northern Mexico with excellent results. Using specific sensors, they calculated tree height, canopy area and number of trees, and with a multispectral camera (PM4), with a resolution of up to 10 cm per pixel, they accurately estimated a number of multispectral indices related to vegetation activity. However, even then, seasonal monitoring is recommended to obtain an accurate estimate of photosynthetic activity and determine the seasonality of plant response. Furthermore, higher-quality mapping requires new research paradigms and the need to adapt algorithms according to forest stand characteristics [72].

### 3.3.2. Low Spatial Resolution and Remote Sensing Signal Anomalies

Remote sensing therefore has great potential in forest monitoring, but most satellites have a low to moderate spatial resolution, which means that a pixel contains a mixture of tree vegetation, undergrowth, soil, shade, etc. [48,49]. This could lead to anomalous index values, particularly in sparse forests and those affected by climate-change-induced mortality [49].

Therefore, it is necessary to estimate the fractional coverage of photosynthetic vegetation, non-photosynthetic vegetation and bare soil. Guerschman et al. [73] developed a very interesting approach, i.e., they used the NDVI and the Cellulose Absorption Index (CAI) to distinguish the different cover types. Analysing large areas of Australia characterised by different cover types (Closed Forest > 80% cover, Non Forest < 20% cover, Open Forest 50%–80% cover and Woodland 21%–50% cover) and using data from the EO-1 Hyperion satellite, with a hyperspectral sensor (30 m spatial resolution), they showed that green vegetation is represented by high NDVI values and an intermediate CAI; dry vegetation and litter by low NDVI values and a high CAI; and bare soil by low NDVI values and a low CAI. In other words, CAI increases linearly with increasing non-photosynthetic vegetation [74]. Furthermore, the work of Guerschman et al. [73] showed that the ratio between the SWIR3 and SWIR 2 bands of MODIS (bands 7 and 6 at 500 m resolution) is linearly correlated with NDVI and CAI derived from Hyperion. Therefore, fractional vegetation cover can be analysed with satellite data (Hyperion and MODIS satellites), but it is still a moderate resolution.

Over time, in order to solve the surface discrimination problem, attempts have been made to reduce the soil signal in the presence of low vegetation cover by adding soil correction factors, resulting in indices such as the Soil-Adjusted Vegetation Index (SAVI) [75], Modified Soil Adjusted Vegetation Index (MSAVI) [76], Optimisation of Soil-Adjusted Vegetation Index (OSAVI) [77] and Generalized Soil-Adjusted Vegetation Index (GSAVI) [78]; alternatively, weighting coefficients were added to improve vegetation signals, as in the case of the indices Enhanced Vegetation Index (EVI) [79], Wide Dynamic Range Vegetation Index (WDRVI) [80] and Near-Infrared Reflectance of terrestrial vegetation (NIRv) [81]. However, these satellite-derived indices are not yet able to accurately capture surface phenological changes due to their limited spatial resolution [49]. In addition, shading causes alterations in indices values, as with NDVI, reducing the accuracy of land cover classification [82].

An approach that could improve this problem could come from comparing NDVI values obtained from satellites, results obtained from radiometers attached to field towers and field data obtained from drones. Wang et al. [49] conducted such an approach in Israel, analysing a *Pinus halepensis Mill.* forest located between the Mediterranean Sea and the Dead Sea, using drones with multispectral cameras with high spatial resolution (around 5 cm at a flight height of 50 m), have improved the accuracy of pine canopy segmentation, vegetation indices and shaded area classification. It was also determined that the satellite data (Landsat 8) were dominated by soil signals (70%), while the tower data were dominated by canopy signals (95%). With these results, discrepancies in NDVI values were recovered and corrected.

Therefore, once again, the use of drones, with the possibility of obtaining high-resolution images, can solve some of the problems encountered by remote sensing with satellites. Of course, in order to use these devices, one must perform a series of systemic time flights over the affected area to obtain an adequate time series. In this way, proximal remote sensing could become increasingly important for forest monitoring, both for the acquisition of remote data and for the calibration/correction of coarser data.

## 4. Conclusions

Studies undertaken so far converge in a single direction characterised by warmer and drier conditions leading to forest dieback and mortality phenomena. The combination of dendrochronology with remote sensing data allow analysing these phenomena from individual trees to global scales. However, this approach presents challenges such as: the decoupling of the NDVI–growth relationship or alterations in remote-sensing indices due to due to mixed pixels and site features. Moreover, considering that the impacts and exposure to climate change are different according to bioclimatic zones and forest types [83,84], the assessment of forest response and vulnerability will also have to be site-specific and

interdisciplinary, with greater caution especially in more heterogeneous regions, such as the Mediterranean basin.

Therefore, to refine the NDVI–growth relationship, it would be useful to analyse the relationships between NDVI and physiological processes at the tree/species level during the growing season in addition to using high-resolution images, such as those obtained from drones, improving the accuracy of the remote sensing indices.

Thus, a multi-proxy analysis could be applied to refine this study, following a cascade flow of qualitative and quantitative information at different scales (in the field and remotely) (Figure 1). Reliable and rapid metrics could be combined to examine large forest areas while preserving local-scale information (proximal remote sensing), along with accurate in situ, dendroecological, physiological (regarding the control of carbon stock, to understand the minimum NSC thresholds required for survival) [7,85] and phenological (concerning the relationships between seasonal tree-ring growth trends and NDVI signal) [56] analyses. In this way, it would be possible to overcome the criticalities of each of the methods used to date and obtain a detailed, large-scale view of forest dieback phenomena. As a result, by improving the understanding of the response of different forest types to climate change, it will be possible to analyse their vulnerability more accurately.

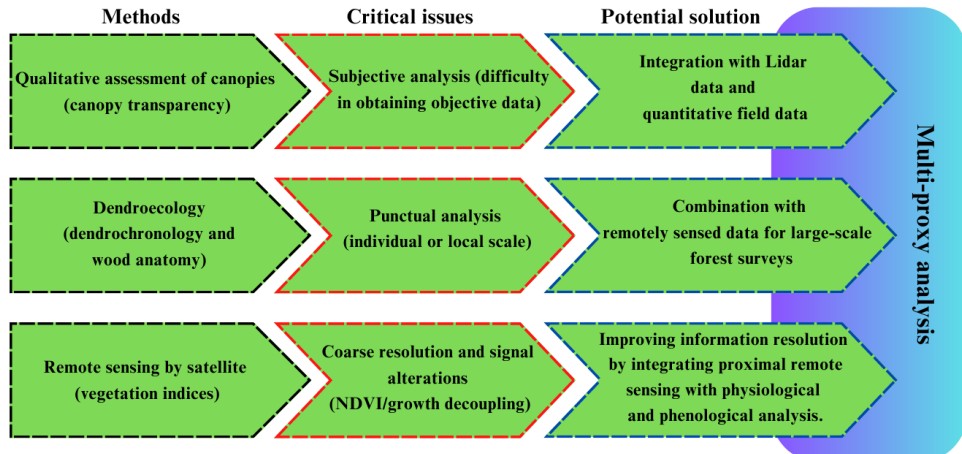

**Figure 1.** Overview summary.

**Author Contributions:** Conceptualization, S.S.P.I. and F.R.; investigation, S.S.P.I., J.J.C. and M.C. (Michele Colangelo); resources, J.J.C. and M.C. (Michele Colangelo); writing—original draft preparation, S.S.P.I.; writing—review and editing, S.S.P.I., J.J.C. and M.C. (Michele Colangelo), F.R., M.P., M.B. and M.C. (Maria Castellaneta); supervision, J.J.C. and F.R. All authors have read and agreed to the published version of the manuscript.

**Funding:** The Italian Ministry of University has supported this research in the framework of the project ARS01_00405, "OT4CLIMA" (D.D. 2261 del 6.9.2018, PON Ramp; I 2014–2020 and FSC). This study was also supported within the Agritech National Research Center and partially financed by the European Union Next-GenerationEU (Piano Nazionale di Ripresa e Resilienza (PNRR)–Missione 4 Componente 2, Investimento 1. 4–D.D. 1032 17 /06 /2022, CN00000022).

**Data Availability Statement:** Not applicable.

**Conflicts of Interest:** The authors declare no conflict of interest.

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
