# Peer review of "Assessing Forest Vulnerability to Climate Change Combining Remote Sensing and Tree-Ring Data: Issues, Needs and Avenues"

_forests, doi:10.3390/f14061138_

Round 1

Reviewer 1 Report (Previous Reviewer 1)

Dear authors.

I read very careful the revised version of your review paper. And in the first review I had mentioned that the work is very qualitatively and interesting. Just needed a little more literature. I saw that you have added more than 20 new papers, which is impressive. I think that you can add in line 86 the following study (with the [29]) (doi.org/10.3390/land11060911), which also reported the dieback of Pinus brutia in Greece and the related consequences in the forest ecosystem. In any case, there are not many studies in relation with forest dieback in Mediterranean, so that can be ignored. I do not have any other comments.

Reviewer 2 Report (Previous Reviewer 2)

I have reviewed and read the attached file in detail and it seems to me that it fully complies with the suggested comments.

The new paragraphs, as well as all the new bibliographical references consulted are appropriate and substantially enrich the article.

This manuscript is a resubmission of an earlier submission. The following is a list of the peer review reports and author responses from that submission.

Round 1

Reviewer 1 Report

Dear Editor.

I have finished my review on the proposed "review" paper “Assessing forest vulnerability to climate change combining remote sensing and tree-ring data: issues, needs and avenues”, forests-2269825-peer-review-v1.

Summary of the manuscript:

In the proposed paper, the author’s goal is to review the current literature, concerning the dieback and mortality phenomena in Mediterranean forest. The review is concentrated mainly in the use of dedroclimatological studies that investigate the effects of climate change on the potential decline of the Mediterranean forest, using also remote sensing data.  

General review:

1. Generally, the manuscript presents an interesting topic and the specific research seems to include some significant points for the research community of this field.

2. The proposed review paper is very well written with very good use of English language. Except some very minor grammatical mistakes and word errors. The author should check again the paper to correct these minor mistakes.

3. The proposed paper is very well structured. It begins with the Introduction with some references that helps the reader to get into the subject immediately. In Introduction there is an effort to provide previous studies with similar scientific content, which took place in the research area and in other countries. Author describes and set very well the scientific problem and how other researchers have approached. At the end of Introduction, authors clearly state the goals of the research. However, I believe that for the specific subject you can enhance the provided literature (see below comments).

4. The methodology of the review is OK.

5. Conclusions are OK.

Additional points for revision:

In my opinion, the proposed paper could be characterized as a very good research work, complies with aims of FORESTS. 

As this is a review paper, it is supposed that the authors should find all the related studies concerning the dieback and mortality phenomena, dendroclimatological studies in relation to climate change in Mediterranean region. In lines 44-51 you report areas that mortality phenomena have recorded. However, you missed some studies from Greece. You should this study (doi.org/10.3390/land11060911), which recorded and described the death of Pinus brutia forest. Also, there are other studies (you should find them) from the same area that analyze the same mass necrosis of pine trees. Also, there is a study (doi.org/10.3390/f13060879), which correlated dendroclimatological data with climate change and found that Abies borisii-regis could potentially withstand the significant increase of temperature in Mediterranean. You should also add this study. The increase of temperature is not always leading to mortality phenomena. There are some other studies that conducted in Mediterranean and found similar results. You should make a better investigation in literature.

Reviewer 2 Report

All suggestions are in the attached file.
